# Corporate Social Responsibility and Social Needs in Health Care Sectors—A Critical Analysis of Social Innovation in the Health Sector in Taiwan

**DOI:** 10.3390/healthcare12151543

**Published:** 2024-08-05

**Authors:** Winnie Chu, Nain-Feng Chu

**Affiliations:** 1Department of Occupational Medicine, Kaohsiung Veterans General Hospital, Kaohsiung 813414, Taiwan; winnie.chu@sciencespo.fr; 2Paris School of International Affairs, Sciences Po Paris, 75007 Paris, France; 3School of Public Health, National Defense Medical Center, Taipei 114201, Taiwan

**Keywords:** social innovation, corporate social responsibility, health sector, COVID-19 pandemic, Taiwan

## Abstract

Background: Social innovation is often used as a mechanism to jump-start public–private partnerships to leverage resources to achieve social impact; the analysis of sustainability and the impact of corporate social responsibility (CSR) cannot be emphasized enough. Due to advances in the information and communication technology industry in Taiwan, this paper aims to explore whether these advancements drive CSR as a form of social innovation to meet health needs in Taiwan. Methodology: This paper uses a case study to look at CSR programs in the health sector in Taiwan. Corporations with diverse missions and different CSR approaches that are available on the internet are selected. The analysis of the case study takes a qualitative, exploratory approach to shed light on current initiatives. Results: The majority of CSR programs in Taiwan are private sector activities that emerged during the COVID-19 pandemic; current CSR activities in Taiwan are driven by awards, public relations, and external interests. Corporations in Taiwan have the potential to address the health care gaps of urban–rural health utilization among Taiwanese indigenous communities. It is recommended for corporations to (1) develop partnerships with public health experts or to (2) employ CSR personnel with health care backgrounds who can navigate the intersection between health, business, and policies to develop CSR strategies. Conclusions: Further evaluation of the projects mentioned in this paper to assess the direct and indirect impact on health outcomes could provide a more comprehensive understanding of the field of CSR in the health sector in Taiwan.

## 1. Introduction

Social innovation, as the OECD defines, is “the design and implementation of new solutions that imply conceptual, process, product, or organizational change, which ultimately aim to improve the welfare and wellbeing of individuals and communities” [1]. Social innovation also takes various forms, from establishing social businesses to corporate innovations. Corporate social responsibility (CSR) is a management concept where companies integrate social and environmental concerns in their business operations and interactions [2]. The COVID-19 pandemic has represented a turning point for the innovation ecosystem. Technology and innovation are expected to enhance health care quality and bring new solutions to bridge health care gaps [3]. Due to the advancement in the Information and Communication Technology (ICT) industry, Taiwan ranks 13th in the World Index of Healthcare Innovation, according to research from The Foundation for Research on Equal Opportunity [4]. However, as social innovation is a fairly new concept in Taiwan, a critical question is whether corporate innovation brings changes that address social needs while being sustainable. This paper first explores the origins of CSR and gaps in the health sector, and then uses Taiwan as a case study to look at CSR programs, whether it achieves the intended goals and results, and concludes with relevant recommendations.

### 1.1. Corporate Social Responsibility

Today’s CSR programs have their roots in corporate philanthropy, where wealthy businessman and philanthropist Andrew Carnegie challenged wealthy people to support social causes. CSR, however, only truly began to take hold in the United States in the 1970s, when the concept of the “social contract” between businesses and society was outlined. Various organizations worldwide now engage in CSR programs to show their corporate commitments and responsibilities toward sustainable development. CSR has evolved from a nice thing to do to a necessity for a successful business. The social, health, and environmental problems faced by global societies currently suggest more responsibility should be taken by actors such as corporations [5]. 

Though originating from generating social impact, CSR received criticisms such as promoting greenwashing or social washing [6]. Corporations may be reluctant to shift their focus from profit-making and shareholders may not want to adopt CSR programs. The study by Wu et al. (2017) indicates that given Asia’s firm culture, firms that are collectively owned or government-owned are more likely to take serious efforts to integrate CSR into aspects of their business operations, while firms that are still expanding and family firms are less likely to devote efforts into CSR activities since the firms’ financial performance has more importance. At the same time, the same study also highlights that firms that devote efforts into CSR are more likely to earn higher earnings and gain a better reputation, which can reduce business risks. This narrative then fits into the speculation that for firms to receive higher earnings they may engage in greenwashing under the name of CSR [7]. 

### 1.2. CSR Is in Dire Need in Health

Although CSR has been a widely known concept in business institutions, there is limited literature about CSR in the health sector. CSR in health can be introduced from inside and outside the health sector. Corporations outside of the health sector can contribute to a better health system or engage in health-promotion activities. Although the practices of health care are already benefiting society and providing services, health care firms ought to uncover the roots of their actions and look at their impact on the beneficiaries [8]. Pharmaceutical companies or health care providers can also go beyond their profit objectives. Since the world was menaced by the stubborn COVID-19 pandemic, there has been a call to strengthen health systems.

CSR for the health sector can occur in many forms; examples of some activities that can be executed include, but are not limited to the following [9]:Develop community partnerships for health promotion and equitable access to health;Provide financial and technical support to advance medical technology and contribute to cutting-edge medical research;Integrate equitable health care delivery through intersectoral partnerships;Manufacture and sell drugs at a lower price for an economically vulnerable population;Protect environmental health through the use of green-energy sources to supply power for medical institutions and manage waste with respect to the environment;Monitor the product supply sourcing of the whole supply chain to guarantee the procurement of materials and equipment from pharmaceutical firms known for the quality of their products that respect the environment, labor laws, and human rights;Support fundraising for rare diseases.

The locations corporations choose to implement CSR activities can be different from the firm’s location. This paper uses Taiwan as a case study to explore social innovations and to consider how corporations have implemented CSR activities in the health sector, given that Taiwan already has a great national health insurance system and will be expecting its fastest economic growth for more than a decade [10]. 

## 2. Methodology

This paper uses a question-driven micro-case study to explore CSR programs in the health sector in Taiwan. The case study attempted to be as comprehensive as possible, and selected different actors in the health ecosystem as part of the analysis. The following questions were the guiding questions and determined the inclusion criteria—(1) Is the mission of the organization clear? (2) Is the idea of the project clear? (3) Was the project time frame mentioned? and (4) Is it clear who were the target group of beneficiaries of the project? 

Based on the information available on the organization’s website or in their annual report, corporations with diverse missions and different CSR approaches are selected as part of the case study. At the end of this selection process, the seven projects that were included in the case study represented health tech, a health care provider (hospital), a medical equipment manufacturer, pharmaceutics, a multinational tech company, nutrition, and a public health foundation. The analysis of the case study uses a qualitative, exploratory analysis to shed light on current CSR initiatives in the health sector and whether they are addressing health care gaps. 

## 3. Case Study

### 3.1. Taiwan’s Health Care System

Taiwan prides itself on its accessible, low overall cost of health care and universal health care coverage. In fact, due to the advancement in the ICT industry, Taiwan ranks 13th in the World Index of Healthcare Innovation, according to research from The Foundation for Research on Equal Opportunity [4]. In the 1980s, the Taiwanese government recruited several U.S health economists for health policy reforms. The group recommended a single-payer model, in which the government was the sole provider of the health insurance, and the financing was modeled on the German system, in which premiums are deducted through individual payroll taxes. Building on the principles of equity in access and benefits, effective and egalitarian cost control, and administrative simplicity to help the public understand the system [11], the national health insurance system (NHI) was implemented in 1995. The NHI benefit package is more comprehensive than any typical Western single-payer system. The coverage not only includes inpatient and outpatient care (preventative, primary, and specialist care), but also includes dental care, mental health care, physical rehabilitation, home nursing, and traditional Chinese medicine. Enrollment in NHI is mandatory for all citizens and foreigners legally residing in Taiwan for longer than six months. Virtually all residents are enrolled. 

In 2009, Taiwan further introduced a universal system of electronic health ID cards, which resemble chip-based credit cards that store patients’ demographic and health information, including past medical visits and histories, claims data, and their prescription file. Health care services can only be provided if a health care professional and a patient simultaneously use their cards to confirm a transaction. Each individual health facility can also use the card to check for past visitations and prescription history at other health facilities to streamline health care provision to the public and health care administration for the professionals. This system gives Taiwan real-time claims data, enabling the government to identify areas of increased utilization, which notably helped Taiwan prevent the initial outbreak of COVID-19 [4].

Health care is an important factor in poverty alleviation. With a national system that is efficient, currently, Taiwan’s public health spending accounts for under 6% of its GDP. However, the health care system is not without flaws. For instance, copayments for outpatient prescription drugs covered under NHI are capped at TWD 200 (USD 6.6) per outpatient visit, regardless of how many drugs are prescribed during that visit; there is no annual cap on drug copayments [11]. While the low cost of care is attractive at first glance, the caveats to this single-payer and low-cost universal coverage can result in an over-reliance on hospital care and over prescription. 

There is also a gap in urban–rural health care utilization. Despite the implementation of NHI covering up to 99% of the population, there is an urban–rural disparity in preventative health care utilization. Studies have suggested that even with NHI, health intervention efforts, such as preventive medicine, might not effectively reach the more rural residents of Taiwan. As a matter of fact, there is a phenomenon of “coverage without access” among the Taiwanese aboriginal population, mostly residing in mountainous townships and experiencing lower socioeconomic status, and exhibiting poorer social determinants of health. They also have limited access to adequate high-quality health care services [12,13].

### 3.2. Emergence of Corporate Social Responsibility as Social Innovation in Taiwan

Renowned for its export-oriented technologically intensive economy, Taiwan also has a long tradition of charitable religious giving. The tradition of charitable religious giving alone created a favorable ecosystem for social innovations to thrive. As a result, social innovations have been blooming in Taiwan over the past two decades. However, an analysis raised the question of the sustainability of social businesses in Taiwan, as the government and charity giving provides quite a generous funding for social enterprises. In 2018 alone, 12 different branches of the government provided over TWD 88 billion over a 5-year period. This high amount of funding may create a dependency on governmental support, rather than seeking other external investors or the development of their business model [14].

As for corporations, similarly to other countries, CSR developed as a part of corporations’ efforts to engage in social innovation and social responsibility, in addition to their profit-driven activities. Current establishments of CSR tend to focus on the rights of employees, shareholders, environment and sustainability community development, and the supply chain relationship. However, there are gaps in the implementation and reporting of CSR.

### 3.3. Existing CSR Projects in Health

In Taiwan, CSR is still dominated by large-scale companies in the financial sector and science and technology industries. Few medical industries and hospitals have published their social sustainability reports. While there are some studies on the lack of CSR performance of medical institutions in Taiwan, among the publicly released social responsibility sustainability reports of the hospitals, the contents mainly focus on the current situation of the hospital, the management, the friendliness of the workplace, and medical services [15]. Even though CSR intends to create a positive social impact, the findings also point out that the CSR of hospitals has a different “quality”. In general, hospitals operated by private universities do poorly in CSR, which is an apparent result of the current laws that do not ask them to abide by rules of statutory public welfare expenditures, whereas medical research centers have more of a tendency to compile reports on CSR [16].

CSR projects in health are either implemented by hospitals as part of their efforts towards contributing towards environmental sustainability or are implemented by other industries to advance the developments in health. The following CSR projects in the health sector in Taiwan stood out:Advancing Medical Technology: AUO, a company that specializes in optoelectronics, partnered with ADLINK Technology, the global leader in industrial computers, to introduce 20 medical panel computers with high-resolution, anti-glare, and touch-screen medical display technology to National Taiwan University Hospital to enhance the infection prevention and control needs during the pandemic. The use of these high-tech displays is expected to safeguard the health of medical personnel in Taiwan, and to upgrade the efficiency and accelerate the digital transformation of the medical industry [17].Integrative approaches for a greener environment: Missioncare, a health care provider and institution, not only adheres to practices that are outlined under good environmental sustainability practices, but also values the rights and potential of its employees. Missioncare offers a scholarship for the employees that attend part-time executive programs and offers free yearly physical exams for all of its employees [18].Donate PPE for those in need: Medtecs, a leading personal protective equipment (PPE) supplier, donated surgical masks, gloves, and many types of PPE during the COVID-19 pandemic to NGOs in Taiwan and some organizations abroad for emergency support. However, Medtecs has only incorporated CSR since February of 2020 and some donation recipients seem to be driven by external factors like diplomatic ties [19].Well-rounded CSR that incorporates different dimensions of health improvement: Acer, a Taiwanese multinational hardware and electronics corporation, has been incorporating CSR to advance social development and sustainable development. However, during the pandemic, they added another category to their CSR under “COVID-19 response”. The newly added activities include an employee blood drive for the community, collaboration with the National Defense Medical Center to use AI technologies for accelerated vaccine development, and collaboration with the Taiwan Centers for Disease Control to establish a “THAS” surveillance system for infection and antimicrobial resistance that helps hospital administrators understand the status of infection and antibiotic resistance in their hospitals. This innovation was awarded the Global ICT Excellence Award—COVID-19 Best Industrial Solution [20].Leverage community partnerships to elevate wellness of the elderly: Pfizer Inc, the American multinational pharmaceutical and biotechnology corporation that developed a COVID-19 vaccine, has a branch in Taiwan. In face of population aging, since 2012, Pfizer Taiwan partnered with local NGOs to form volunteer groups to keep senior citizens who live on their own and in remote areas company during the holiday seasons and took them for holiday shopping [21].Health promotion and nutrient consumption matters: Taiyen Biotech is a salt manufacturer and actor in customer-oriented service industry that safeguards public health and beauty with premium biotech products. In its 60-year history, the company has devoted its efforts to public health, promoting the consumption of sodium and potassium [22].Creating a Healthy and Happy Tomorrow through Medicine and Nutritional Care: Taiwan Millennium Health Foundation, a foundation created as part of the CSR initiatives of Uni-President, the largest food production company in Taiwan and Asia, aims to promote healthy lifestyles and medical knowledge through health campaigns and its “National Health Day” to screen for metabolic diseases in Taiwan. Together with 7-eleven, the leading convenience store in Taiwan, they supply blood pressure monitors, waistline tape measures, and pamphlets for health education to raise awareness of metabolic diseases and the benefits of maintaining a healthy lifestyle [23].

## 4. Discussion

The projects above are initiatives of multinational companies, health care providers, or local Taiwanese firms, and these projects encompass a great variability of what CSR in health could be. While some larger firms in Taiwan have been incorporating CSR since the early 2000s, many CSR projects only started under the wider mission of creating a better environment and improved employee wellbeing after the Company Act came into force under the Ministry of Economic Affairs in 2018. This act writes the following: “when conducting its business, every company shall comply with the laws and regulations as well as business ethics and may take actions which will promote public interests in order to fulfill its social responsibilities” [24]. A number of health projects only emerged during the COVID-19 pandemic through mechanisms of in-country donations and “PPE-diplomacy” to Taiwan’s allies in Southeast Asia (i.e., Medtecs). These approaches have good intentions but only generate a time-bound effect, which contradicts the purposes of social innovation for sustainable change and can equally be scrutinized as engaging in social washing.

To promote innovation in the private sector, the government awards the best CSR programs. Since Taiwan is an export-oriented economy and CSR is now increasingly used as part of company branding and strategy to gain momentum for international collaborations, many companies are forced to engage in CSR activities or else they risk losing contracts. Substantial evidence also suggests that CSR and financial earnings are intertwined; corporations are self-serving and only engage in CSR when it yields benefits. Typically, charitable giving attracts the publics’ attention, so the media only writes about big fundraisers and donations while CSR programs on community engagement and sustainability do not grab the interest of the public. This resulting external pressure, in turn, affects how corporations engage in CSR. While the CSR projects in the health sector illustrated above show how diverse CSR programs can be, it also reflects the trends in public interest. Donations for medical devices have always existed and PPE donations gained momentum during the COVID-19 pandemic, yet none of these CSR programs address the need of closing the urban–rural health utilization among Taiwanese indigenous communities.

With gaps to fill, CSR is only at the margins in Taiwan. Just like boosting social innovation, effective and sustainable CSR requires talent and cross-sector collaboration to reach full potential. Current CSR activities in Taiwan are driven by awards, public relations interest, and other external interests. As the type of CSR activities in Taiwan that attract media attention are philanthropic activities, it is also suspected that CSR positions in corporations will not require much technical expertise on sustainable development. A review of several recruiting announcements for CSR positions on 104.com, a popular job board in Taiwan, confirmed these speculations. With the exceptions of companies already working in the environmental or biomedical field that specifically require their CSR personnel to have relevant knowledge and experience in sustainable development, climate change and/or carbon reduction, CSR positions in other firms do not have a targeted profile and are generally positioned under the offices of public relations, administration, or donor relations. Positioning CSR under offices of donor relations or public relations will eventually cause CSR strategies to deviate from their intended impact.

## 5. Conclusions

There is still room for improvement in CSR programs to address health needs in Taiwan. CSR in Taiwan promotes SDG washing. Big corporations are under scrutiny for engaging in CSR greenwashing and SDG washing, as corporations often face pressure to engage in philanthropic activities for media interests, which aligns with the current landscape of the risks of CSR. Engaging in these activities will only undermine the importance of CSR and SDGs in the long run. To make the whole CSR system work, it is recommended for management to take a participatory approach to enable a business ecosystem that is ready to create changes.

Findings from this paper also suggest that CSR strategies also often come from a top-down approach, and firms in Taiwan have strong hierarchical culture; hence, structural and strategy change within corporations is needed to align CSR back with its initial goals to bring social impact. To fill the current health care gaps, it is also recommended for corporations to either develop partnerships with public health experts or employ CSR personnel with health backgrounds who are able to navigate the playing field between health, business, and policies to develop CSR strategies. However, there are some limitations to this study. While the case studies shed light on existing efforts in CSR in Taiwan, these are hindered by the quantity of CSR projects in the health sector and the availability of evaluations on these projects; future research on the approach of CSR health projects initiated after the COVID-19 pandemic and an evaluation on the projects mentioned in this paper to assess the direct and indirect impact on health outcomes could provide a more comprehensive understanding of the field of CSR in the health sector in Taiwan.

## Data Availability

Not applicable.

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
