# Peer review of "Corporate Social Responsibility and Social Needs in Health Care Sectors—A Critical Analysis of Social Innovation in the Health Sector in Taiwan"

_healthcare, 2024, doi:10.3390/healthcare12151543_

Round 1
Reviewer 1 Report
Comments and Suggestions for Authors
Comments are in the attached file. The paper presents an interesting issue but needs significant improvements.

Some little spelling errors were found, namely repetition of words ("that that"), but the language is good and understandable.
Author Response
Response to Reviewer Comments
Overall comments: This manuscript presents an interesting issue for healthcare. It focuses on one specific context, that is Taiwan, and tries to explore it as a case study. However, it tends to be more likely a reflection and/or an opinion article than a research report, because the structure of the paper has some weaknesses. The methodology and results are not according to the expected for scientific papers.
Abstract: presents a synthesis of the whole manuscript. The study aims and the methods are missing.
We had revised and added methods and results in our update manuscript. Thank you for your comment.
Introduction: presents the topic, the context, and the problem. Some concepts are specified. However, on 1.1. the subtitle is Corporate Social Innovation and Corporate Social Responsibility, but the content of this point only refers to the second concept. Nothing is mentioned about Corporate Social Innovation. The abbreviation ICT is used without clarification.
We had revised in our update manuscript. ICT is the Information Communication Technology. Thank you for your comments.
The Methodology is very short and limited in point 2, mentioning that it is a case study on CSR programs. Following, point 3 is entitled “Case Study”, but it is not clear what this section means. Is it a contextualization of the case study? Are these results of programs analyzed? The next topics don’t seem in line with the academic structure of an article. Some programs are listed, but the analysis techniques are not explicit. Is it content analysis, or documental analysis? What criteria were selected to analyze the programs?
We had revised the sections of methods and case study as results in our update manuscript. Thanks.
Results and Discussion are not clear. After, as the 4 th point, we can see “Conclusions” …. Nevertheless, what is before needs to be aligned for a better understanding of the content and aims of the paper. Even having in mind that it is a “Critical Analysis”.
We had revised the sections of discussion and conclusion in our update manuscript. Thanks.
References are also confusing because the notes in the text and the reference list have different formats and something is wrong, with duplication of the list.
We had revised the order of references deleted the footnote in our update manuscript. Thank you for your comment.
Thus, instead of an interesting topic and focus on a particular context of health care, many weaknesses were found and the text needs big improvements.
We had revised and marked in our update manuscript. Thank you for your comment.
Reviewer 2 Report
Comments and Suggestions for Authors
This is a very interesting paper addressing an issue that is of relevance not only on national/regional level but also with a global perspective. An area for improvement is the description of the methodology used for the case study. How where relevant initiatives identifies, and by which protocol was the analysis of the identified initiatives led, was it cross-checked in collaboration with several researchers, was there a consensus procedure for the final description etc. This would be needed to put the findings in perspective.
Author Response
We had revised the sections of methods and case study as results in our update manuscript. Thanks.
We had revised the sections of discussion and conclusions in our update manuscript. Thanks.
Reviewer 3 Report
Comments and Suggestions for Authors
The paper addresses a topic of great interest and relevance, the innovation capacity of welfare systems, and it does so by delving into research on a particularly interesting and dynamic part of the world, Taiwan. However, it presents limitations that in its current stage of development do not make it publishable and require further evaluation work.
1) First, the paper puts on the same level two concepts that are both influential and current but have different roots, histories and meanings, namely social innovation and corporate social responsibility. Furthermore, after a brief mention, social innovation seems to be forgotten. It is necessary to better frame this part of the contribution and develop the two concepts more thoroughly.
2) The authors introduce the concept of the innovation ecosystem, which has a wide theoretical and bibliographic reference framework, which however is neglected here. It is necessary to deepen this aspect or to avoid citing it.
3) The third and most problematic aspect of the work relates to the methodology. The authors mention a critical analysis of the data presented, but do not explain how this analysis is carried out, how the data is treated, how it was collected, nor on the basis of which theoretical framework it is analysed. In light of this, the paper does not present a clear research process that can be replicated by others
The paper would benefit from:
a) Clearly distinguishing between social innovation and corporate social responsibility, and developing both concepts in more depth
b) Providing a more thorough review of the theoretical framework around innovation ecosystems if this concept is to be included
c) Providing a detailed explanation of the data collection methods used, including
d) Describing the specific techniques and processes used to critically analyse the data
Without these key methodological details, the paper lacks transparency and makes it difficult to evaluate the validity and reliability of the findings. Addressing these issues around the research process would strengthen the overall quality and contribution of the work.
Author Response
The paper addresses a topic of great interest and relevance, the innovation capacity of welfare systems, and it does so by delving into research on a particularly interesting and dynamic part of the world, Taiwan. However, it presents limitations that in its current stage of development do not make it publishable and require further evaluation work.
- First, the paper puts on the same level two concepts that are both influential and current but have different roots, histories and meanings, namely social innovation and corporate social responsibility. Furthermore, after a brief mention, social innovation seems to be forgotten. It is necessary to better frame this part of the contribution and develop the two concepts more thoroughly.
Thank you for your comments. We had revised the introduction section in our update manuscript.
- The authors introduce the concept of the innovation ecosystem, which has a wide theoretical and bibliographic reference framework, which however is neglected here. It is necessary to deepen this aspect or to avoid citing it.
Thank you for your comments. We had revised in our update manuscript.
3) The third and most problematic aspect of the work relates to the methodology. The authors mention a critical analysis of the data presented, but do not explain how this analysis is carried out, how the data is treated, how it was collected, nor on the basis of which theoretical framework it is analysed. In light of this, the paper does not present a clear research process that can be replicated by others
We had revised the sections of methods in our update manuscript. We described that the study is qualitative study design using case study to discuss the results. Thanks.
Round 2
Reviewer 1 Report
Comments and Suggestions for Authors
Some improvements were made, but not so significant. The abstract has now more details about the aim, the methods, and the analysis, as well as some results. Given the limited words for the abstract, if these components were added, the next (conclusions and recommendations) need to be shortened.
The methodology was not improved. The case study has some characteristics and it needs to be clarified based on theoretical framework. If it is a review paper, the authors should clarify it, and say what type of review it is.
A better structure of the text was tried, namely replacing some subtitles, which facilitates the understanding of the article. Nevertheless, the seven projects reported could be analyzed with specific criteria defined and applied to all, to get a stronger research review and critical analysis.
Some references need to be reviewed because in some cases the date in the text is not coherent with the date in the references section (for example Wu et al (2017) versus Wu, S., Lin, F. & Wu, C. (2012)). In addition, the references are not written according to the MDPI format, and it is recommended. Also, citations need to indicate the page, and it is not shown.
Not all the changes are highlighted and it implies more time and attention to the second review. To improve the reviewer’s help and benefit for authors it is highly recommended.
In sum, the paper continues to need so improvements.
Author Response
Reviewer-1
Some improvements were made, but not so significant. The abstract has now more details about the aim, the methods, and the analysis, as well as some results. Given the limited words for the abstract, if these components were added, the next (conclusions and recommendations) need to be shortened.
Thank you for your comments. We revised in the update manuscript.
The methodology was not improved. The case study has some characteristics and it needs to be clarified based on theoretical framework. If it is a review paper, the authors should clarify it, and say what type of review it is.
Thank you for your comments. We revised in the update manuscript. We added following the section of methodology. -- This paper uses a question-driven micro-case study to explore CSR programs in the health sector in Taiwan. The case study attempted to be as comprehensive as possible, and selected different actors in the health ecosystem as part of the analysis. The following questions were guiding questions and determined the inclusion criteria- 1) Is the mission of the organization clear? 2) Is the idea of the project clear? 3) Was the project time frame mentioned? and 4) Is it clear who were the target group of beneficiaries of the project?
A better structure of the text was tried, namely replacing some subtitles, which facilitates the understanding of the article. Nevertheless, the seven projects reported could be analyzed with specific criteria defined and applied to all, to get a stronger research review and critical analysis.
Thank you for your comments. We revised the structure in the update manuscript.
Some references need to be reviewed because in some cases the date in the text is not coherent with the date in the references section (for example Wu et al (2017) versus Wu, S., Lin, F. & Wu, C. (2012)). In addition, the references are not written according to the MDPI format, and it is recommended. Also, citations need to indicate the page, and it is not shown.
We revised the format of references and corrected as (2012) in the update manuscript. Thanks.
Not all the changes are highlighted and it implies more time and attention to the second review. To improve the reviewer’s help and benefit for authors it is highly recommended.
In sum, the paper continues to need so improvements.
Thank you for your comments. We revised and highlighted the changes in the update manuscript.
Reviewer 2 Report
Comments and Suggestions for Authors
Many thanks to the authors for amendments made. However, a paragraph on methodology as asked for I did not find. It should be made clear how the case study methodolgically was implemented, e.g. how the named inititatives were identified and selected.
Recommendation therefor stays the same. Please add a paragraph on methodology
Author Response
Reviewer-2
Many thanks to the authors for amendments made. However, a paragraph on methodology as asked for I did not find. It should be made clear how the case study methodolgically was implemented, e.g. how the named inititatives were identified and selected.
Recommendation therefor stays the same. Please add a paragraph on methodology
Thank you for your comments. We revised and added more information on the methodology section in the update manuscript.
This paper uses a question-driven micro-case study to explore CSR programs in the health sector in Taiwan. The case study attempted to be as comprehensive as possible, and selected different actors in the health ecosystem as part of the analysis. The following questions were guiding questions and determined the inclusion criteria- 1) Is the mission of the organization clear? 2) Is the idea of the project clear? 3) Was the project time frame mentioned? and 4) Is it clear who were the target group of beneficiaries of the project?
Based on information available on the organization’s website or annual report, corporations with diverse missions and different CSR approaches are selected as part of the case study. At the end the seven projects included in the case study represent health tech, healthcare provider (hospital), medical equipment, pharmaceutical, tech multinational company, nutrition, and public health foundation. The analysis of the case study is qualitative as an exploratory analysis to shed light on current CSR initiatives in the health sector and whether it is addressing health care gaps.
Reviewer 3 Report
Comments and Suggestions for Authors
Dear Authors, at the moment I do not consider the revisions of the third point (the metdhodological one) good enough to solve the problems I highlighted in my previous assessment. By affirming 'the work is qualitative' is a too much generic affirmation. I suggest considering the posisbility to explain why and how the case has been choosen, how you have collected data, and especially how you have analyze data and so on.
Author Response
Reviewer-3
Dear Authors, at the moment I do not consider the revisions of the third point (the metdhodological one) good enough to solve the problems I highlighted in my previous assessment. By affirming 'the work is qualitative' is a too much generic affirmation. I suggest considering the posisbility to explain why and how the case has been choosen, how you have collected data, and especially how you have analyze data and so on.
Thank you for your comments. We revised and added more information on the methodology section in the update manuscript.
This paper uses a question-driven micro-case study to explore CSR programs in the health sector in Taiwan. The case study attempted to be as comprehensive as possible, and selected different actors in the health ecosystem as part of the analysis. The following questions were guiding questions and determined the inclusion criteria- 1) Is the mission of the organization clear? 2) Is the idea of the project clear? 3) Was the project time frame mentioned? and 4) Is it clear who were the target group of beneficiaries of the project?
Based on information available on the organization’s website or annual report, corporations with diverse missions and different CSR approaches are selected as part of the case study. At the end the seven projects included in the case study represent health tech, healthcare provider (hospital), medical equipment, pharmaceutical, tech multinational company, nutrition, and public health foundation. The analysis of the case study is qualitative as an exploratory analysis to shed light on current CSR initiatives in the health sector and whether it is addressing health care gaps.
Round 3
Reviewer 2 Report
Comments and Suggestions for Authors
Thanks much to the authors for adding a paragraph on methodology. I am sorry I could not sufficiently make clear my comment/request. What would be important to know is HOW the 7 organisations were selected for further analysis,. Can you please add a sentence answering this question.
Author Response
Reviewer-2
Many thanks to the authors for amendments made. However, a paragraph on methodology as asked for I did not find. It should be made clear how the case study methodolgically was implemented, e.g. how the named inititatives were identified and selected.
Recommendation therefor stays the same. Please add a paragraph on methodology
Thank you for your comments. We revised and added more information on the methodology section in the update manuscript.
This paper uses a question-driven micro-case study to explore CSR programs in the health sector in Taiwan. The case study attempted to be as comprehensive as possible, and selected different actors in the health ecosystem as part of the analysis. The following questions were guiding questions and determined the inclusion criteria- 1) Is the mission of the organization clear? 2) Is the idea of the project clear? 3) Was the project time frame mentioned? and 4) Is it clear who were the target group of beneficiaries of the project?
What would be important to know is HOW the 7 organisations were selected for further analysis,.
Based on information available on the organization’s website or annual report, corporations with diverse missions and different CSR approaches are selected
study represent health tech, healthcare provider (hospital), medical equipment, pharmaceutical, tech multinational company, nutrition, and public health foundation. The analysis of the case study is qualitative as an exploratory analysis to shed light on current CSR initiatives in the health sector and whether it is addressing health care gaps.
Reviewer 3 Report
Comments and Suggestions for Authors
Dear Authors, this new version of the paper includes a methodology section clear enough to consider the paper publishable, and the rest of the document had been previously reviewed
Author Response
Reviewer-3
Dear Authors, this new version of the paper includes a methodology section clear enough to consider the paper publishable, and the rest of the document had been previously reviewed.
Thank you for your comments.
Dear Authors, at the moment I do not consider the revisions of the third point (the metdhodological one) good enough to solve the problems I highlighted in my previous assessment. By affirming 'the work is qualitative' is a too much generic affirmation. I suggest considering the posisbility to explain why and how the case has been choosen, how you have collected data, and especially how you have analyze data and so on.
Thank you for your comments. We revised and added more information on the methodology section in the update manuscript.
This paper uses a question-driven micro-case study to explore CSR programs in the health sector in Taiwan. The case study attempted to be as comprehensive as possible, and selected different actors in the health ecosystem as part of the analysis. The following questions were guiding questions and determined the inclusion criteria- 1) Is the mission of the organization clear? 2) Is the idea of the project clear? 3) Was the project time frame mentioned? and 4) Is it clear who were the target group of beneficiaries of the project?
Based on information available on the organization’s website or annual report, corporations with diverse missions and different CSR approaches are selected as part of the case study. At the end the seven projects included in the case study represent health tech, healthcare provider (hospital), medical equipment, pharmaceutical, tech multinational company, nutrition, and public health foundation. The analysis of the case study is qualitative as an exploratory analysis to shed light on current CSR initiatives in the health sector and whether it is addressing health care gaps.